# Regional Body Composition and Strength, Not Total Body Composition, Are Determinants of Performance in Climbers

**DOI:** 10.3390/jfmk9040228

**Published:** 2024-11-12

**Authors:** Fernando Carrasco, Maria Jose Arias-Tellez, Ignacio Solar-Altamirano, Jorge Inostroza, Gabriela Carrasco

**Affiliations:** 1Department of Nutrition, Faculty of Medicine, University of Chile, Independence 1027, Santiago 8380000, Chile; fernandocarrasco@u.uchile.cl (F.C.); mariajosearias@uchile.cl (M.J.A.-T.); jinostrozae@med.uchile.cl (J.I.); 2Department of Physical Education and Sports, Faculty of Sport Sciences, Sport and Health University Research Institute (iMUDS), University of Granada, 18071 Granada, Spain; 3Free Sport Foundation, Los Acantos 1110 Depto. 102, Santiago 7550000, Chile; ignacio@deportelibre.cl

**Keywords:** climbers, body composition, handgrip strength, dual-energy X-Ray absorptiometry, athletic performance

## Abstract

**Objective:** To compare the body composition of Chilean climbers of different performance levels and to determine the relation between the forearm and upper-trunk lean mass and the handgrip and upper-body traction strength, respectively. **Methods:** A cross-sectional study was carried out on thirty Chilean male adult climbers (26.1 ± 4.9 y.): nine of intermediate level (L1), eleven advanced (L2), and ten elite (L3). Through dual-energy X-Ray absorptiometry (DXA; Lunar Prodigy^®^), fat mass percentage (FM%), total lean mass (LM), forearm lean mass (FLM), and upper-trunk lean mass (UTLM) were measured. Total muscle mass (TMM) was also estimated. Handgrip strength (HGS) was measured with a Jamar^®^ dynamometer. Maximum upper-body traction strength (UBTS) was evaluated with a standardized movement. The level of climbing was assessed according to IRCRA rules. **Results:** No differences in FM%, total LM, UTLM, or TMM between the groups were found. Left and assistant FLM were significantly higher in L3 (*p* = 0.047 and 0.041, respectively). HGS absolute, relative, and adjusted by FLM were not different between groups. FLM was associated with HGS in all segments (*p* ≤ 0.001). UBTS absolute values, and as adjusted by TMM, were significantly higher in L3 (*p* = 0.047 and *p* = 0.049, respectively). **Conclusions:** Left and non-dominant forearm lean mass were significantly higher in elite climbers. Handgrip strength was not significantly higher in elite climbers; however, the upper-body traction strength was significantly higher in elite climbers, independent of total or regional muscle mass.

## 1. Introduction

Sport climbing has been gaining popularity around the world [1,2]. Climbing sports encompass a variety of disciplines [3], each with its own unique techniques, equipment, and environments. Broadly, these include indoor wall climbing and outdoor rock-climbing, with several sub-disciplines under each, activities in which physiological aspects of athletes have been related to better sport rock-climbing performance [4,5,6,7,8]. In particular, previous studies have made apparent the importance of having higher strength and endurance in forearms, hands, and fingers, together with a lower total body fat percentage, in this discipline [9,10,11,12,13]. However, whether forearm and upper-body composition are determinants of climbing athletic performance is still not clear.

Traditional measurements through hand and finger dynamometry have been compared between climbers and non-climbers [14,15,16]. Interestingly, Cutts et al. were the first to show that high-performance rock climbers have significantly greater whole-handgrip strength (HGS) and pinch grip strength than non-climbing individuals of the same sex and physical condition [17]. Similarly, analyzing results between elite climbers, Watts et al. showed that elite sport climbers have a higher HGS-to-body mass ratio when compared with other athletic groups [18]. More recently, Ozimek et al. have found that elite climbers recorded significantly higher values for finger strength than did advanced climbers [19]. However, MacLeod et al. also have shown that despite intermediate rock-climbers having a higher mean maximum voluntary contraction in the fingers, they do not demonstrate differences in endurance tests compared with non-climbers; the study concluded that other physiologic variables such as muscle re-oxygenation during rest phases could be a predictor of endurance performance [11].

The total body composition has been studied in this discipline, highlighting that a lower fat mass percentage, together with a higher HGS, would be directly related to better performance [9,12,13,20]. Nowadays, studies have proposed that forearm/HGS could be an important factor in maintaining grip while climbing and demonstrating a high realtioship with performance [8,11,21]. In this line, Fanchini et al. show that boulderers have a higher grip strength (relative to total body mass) than non-climbers and lead climbers [22]. Indeed, Fryer et al. showed that boulderers had greater forearm strength than lead climbers and the control group, even after adjustments for body mass, which is likely explained by neural adaptation, rather than hypertrophy, reflecting the specific adaptations that determine performance among different rock-climbing disciplines [10]. On the other hand, previous studies have reported that higher strength and endurance during upper-body tests are related to climbing performance levels [12,23,24]. Better conditioning of arms, shoulders, and upper torso could be advantageous due to the increased demand required by higher-level climbing problems, particularly those relating to an overhanging profile.

In addition to these research lines, the most recent studies have shown that not only strength but also the rates of force development of the finger flexor [25] and upper body [26] show distinctions among performance levels. However, the evidence connecting forearm and upper-body lean mass composition and strength in the climbing population, as well as any exploration of how these factors might relate to sports performance, is lacking.

Therefore, the aims of this study were to compare the total and regional body composition, evaluated by DXA, in Chilean climbers of varying performance levels and to examine the relationship between the forearm and upper-trunk lean mass and the handgrip strength and upper-body traction strength, respectively.

## 2. Materials and Methods

### 2.1. Study Design

A cross-sectional investigation in a sample of 30 Chilean climbers, healthy male adults (age 26.1 ± 4.1 y.; BMI 22.1 ± 1.5 kg/m^2^), with 9 being of intermediate level (L1), 11 advanced (L2), and 10 elite (L3), all with climbing practice of at least three months, was carried out. The recruitment of volunteers was made by convenience sampling. The climbing level was assessed by self-report according to the recommendations of the International Rock Climbing Research Association [27]. This method follows the rule 3:3:3 and consists in the recording, during the participants interview, of the highest redpoint grade for which they have completed three successful ascents on three different routes (at the relevant grade) within the previous three months. Taking into account this grade, the climbing level of the climber was categorized following the cut-off points of the IRCRA detailed in the reference. For example, following a climber declaring the following achievements in the last three months, ten different routes were graded 5.10d; five different routes were graded 5.11c; three different routes were graded 5.12a; and two different routes were graded 5.12c. The relevant climbers’ grade considered for the primary categorization is 5.12a. Following the cut-off points of IRCRA, classifies the climbers as “advanced”.

The study was approved by the Ethics Committee of the Faculty of Medicine of the University of Chile (project n° 217–2016) and met all the requirements of the Declaration of Helsinki and the Ethical Standards for Research in Sports and Exercise. All volunteers signed an informed consent prior to their participation. The measurements were made at the Department of Nutrition of the Faculty of Medicine, University of Chile, between January and April 2017.

### 2.2. Procedures

#### 2.2.1. Anthropometry and Body Composition

Weight (kg) and height (m) were measured using a calibrated digital scale SECA (model gmbh & Co. Hammer Steindamm 3–25, Hamburg, Germany) and a stadiometer brand SECA (model 220), respectively. BMI (kg/m^2^) was calculated.

Body composition was evaluated with double-energy X-Ray absorptiometry (DXA) with Lunar Prodigy Advance equipment (General Electric Systems, Madison, WI, USA, v15 SP2 software), obtaining as quantitative results the bone mineral content (BMC), total fat mass (FM), total lean mass (LM), and appendicular lean mass (ALM) [28]. In addition, the forearm and upper body were established as regions of interest (ROI) through use of the tool for segmental analysis of subregions, obtaining forearm lean mass (FLM) and upper-body lean mass (UBLM) [28,29]. In addition, total muscle mass (TMM) was estimated using a prediction model based on the summation of ALM measured by DEXA, as validated against magnetic resonance imaging: TMM, kg = (ALM, kg × 1.19) − 1.65 (adjusted R^2^ = 0.96; SE = 1.46 kg) [30].

#### 2.2.2. Strength Assessment

HGS was measured with a calibrated Jamar^®^ hydraulic dynamometer (Sammons Preston, Bolingbrook, IL, USA), using the standardized protocol of the American Society of Hand Therapists [31]. The maximum strength achieved after 3 attempts was evaluated according to the recommendations of Savva et al. [32], with 1 min rest intervals between each attempt, for each upper limb. Both laterality and hand dominance were recorded for each recorded maximum measurement. HGS relative to total body weight (RTW) was calculated and adjusted for TMM and FLM.

The maximum upper-body traction strength (UBTS) produced against an external load in addition to its own body weight was evaluated during a standardized movement [33] developed to calculate 1 maximum repetition (1-MR-UBTS). The UBTS test followed specific characteristics and protocols [34]: (i) the equipment consisted of a Metolius hang board on a 10° overhanging wall, harness, and free weights; (ii) the valid test conditions comprised, for the hand, a positioning-full-hand grip on the hang board, and for the upper-body, a pull-up—pull-up executed with symmetrical shoulder elevation and chin above the hand-hold. Both conditions were supervised to validate the attempt. The 1-MR-UBTS RTW was calculated and adjusted for TMM and UBLM.

### 2.3. Statistical Analysis

Analyses of descriptive statistics were performed to examine the characteristics of the study participants. The distribution of the variables was verified using the Shapiro–Wilk test, skewness and kurtosis values, a visual check of histograms, Q-Q, and box plots, with conclusions of normality determined for all variables. ANOVA was used to assess differences between groups as to athletic performance. A Bonferroni post hoc test was applied for multiple comparisons. A partial correlation coefficient was computed to adjust for the potential effect of forearm lean mass on HGS. Pearson’s correlation-coefficient analyses were performed to evaluate the associations of the FLM and UBLM (total, relative to body mass, and adjusted by TMM and body mass) with the HGS and UBTS, respectively. SPSS 21.0 statistical software was used for statistical analysis (SPSS, Inc., Chicago, IL, USA). Significance was defined as a *p* value < 0.05.

## 3. Results

Table 1 shows the characteristics and body composition of the climbers. There were no differences between the groups as to FM%, LM, UBLM, or TMM. The left and assistant FLM were significantly higher in L3 climbers compared to the L1 group (*p* = 0.047 and 0.041, respectively).

Table 2 summarizes the measurements of hand grip and upper-body strength in the three performance levels of the climbers. There was no difference between groups in handgrip strength, both as to absolute values and as adjusted for ipsilateral forearm lean mass. The mean 1-MR-UBTS was significantly higher in L3 than in the L1 group, when expressed in absolute values (*p* = 0.047) and adjusted by total muscle mass (*p* = 0.049), but not when expressed in relation to total body mass (*p* = 0.202).

Figure 1 shows that forearm lean mass was associated with HGS in the right (r = 0.603), left (r = 0.587), dominant (r = 0.571), and assistant forearms (r = 0.624), all with *p* < 0.001.

Figure 2 shows no significant correlation between 1-RM-UBTS and UBLM, both as absolute (*p* = 0.982) and as expressed in RTW (*p* = 0.265). In addition, no significant correlation was observed between 1-RM-UBTS and TMM (r = 0.076; *p* = 0.690) or UBLM (r = 0.004; *p* = 0.982).

## 4. Discussion

The present results show that left and non-dominant forearm lean mass were significantly higher in elite-level climbers compared to intermediate climbers. Though handgrip strength and upper-trunk traction forces were not significantly higher in elite climbers when they were expressed in relation to body weight, handgrip strength was significantly associated with forearm lean mass.

Our findings showed no differences in FM% among the groups. According to published scientific evidence, climbers associated with higher sports performance are characterized by showing lower FM% values, compared to climbers associated with lower performance [9,12,13,19,20]. However, this characteristic does not agree with the values observed by Grant et al. [24] and the findings of the present study. This discrepancy could, in part, be related to the methods and estimation formulas used to assess the body composition [35]. Skinfold measurement and bioelectrical impedance analysis have shown significant differences relative to DXA in athlete populations [36]. Among climbers, these methods have been shown to underestimate FM% [37]. The few studies using DXA have compared climbers of a similar level of performance or compared climbers to non-climbers [35,38], making it difficult to establish a contrast with our results.

Regarding TMM, previous studies have not quantified TMM or regional LM, due to the impossibility of predicting them with skinfolds or with bioimpedance analysis. In this sense, the application of DXA makes it possible to estimate TMM with a high predictive value as compared with magnetic resonance imaging measurements (gold-standard method) [30,39]. In addition to this, DXA allows the measurement of body composition in regions of interest, such as the limbs, where LM is mainly constituted by muscle mass [28]. Due to this advantage, DXA measurements have been incorporated in the diagnosis of low muscle mass in the definition of sarcopenia [40], but to date, it has not been used to its full potential in the evaluation of athletes [41]. In the present study, TMM was not higher in elite climbers, but there was a trend towards greater FLM on the dominant side and significantly higher FLM on the assistant side. In addition, FLM had a positive and significant correlation with ipsilateral hand-grip strength.

HGS has been proposed as a determinant of performance in climbing, especially when it is expressed as a percentage of total body weight [5,12]. Different studies have shown an association between higher HGS and better performance scores in elite climbers [8,9,24] or for climbers vs. non-climbers [17,24]. In contrast, we observed that elite climbers did not present an HGS significantly higher than did climbers of an advanced or intermediate performance level. Part of this discrepancy could be explained by the type of dynamometer used, although in three studies, the same instrument (Jamar^®^) was used [5,9,12], one which has been validated in healthy university students [31]. Even so, among the limitations relevant to the application of this dynamometer is the need for adjustment, which has been described relative to the length of the hand [42]. Although, with the Jamar^®^ device, it is possible to adjust the grip length between five levels (with a fixed difference of 1/2 inch between each level), the protocols perform measurements only in the intermediate position, as applied in our study. Validity and reliability studies of different dynamometers have shown that those of the Takei KK type, such as the one used by Grant et al. [24], would be more suitable for this type of athlete [43]. Another aspect to consider could be the type of upper-limb muscle work during climbing and the best way to measure it. In climbing, the demand is for isolated isometric contraction of the flexor muscles of the fingers, and not of the pincer or grip type, such as that measured by handgrip dynamometry, so this would not be a specific tool for this sport [44]. Even with these limitations, there is agreement that hand dynamometry can be used as a complementary tool for the evaluation of these types of athletes. Taken together, these results put focus on the development of adequate muscle hypertrophy of the forearm, particularly of the left and non-assistant side, as a component of high-level climbing performance. Specific measurement devices, such as dynamometers fixed to a climbing rung, could better discriminate with respect to the influence of finger/hand strength or rate of force development in the context of a specific climbing exercise.

Regarding upper-body traction strength, significant differences between the intermediate and elite levels of performance were found when expressed in absolute values or adjusted by total muscle mass. The lack of association of UBTS with UBLM or TMM could be explained in part by the following: (i) the ROI selection, leaving aside appendicular muscle mass that participates in the exercise; and (ii) a higher relevance of other factors, such as neural activation or muscular coordination, in producing higher levels of force. Considering those possibilities, development of the superior upper-body strength [9], rate of force development [45], and/or force-velocity profile [46] of elite-level climbers should incorporate training which focuses on factors beyond muscle hypertrophy.

Some strengths of the present study must be highlighted. The use of a reference method for the estimation and comparison of body composition, the inclusion of three distinct performance-level populations, and the register of the athletes under the IRCRA classification and grouping statement allow comparisons to be made in future studies. On the other hand, we acknowledge that some limitations, when interpreting and comparing the results, could include the following: (i) the absence of female athletes; and (ii) the limited number of subjects in each group as defined by climbing performance.

In conclusion, the lack of association between FM% and performance level in the present study contrasts with much of the published scientific evidence, although part of the discrepancy could be related to the methodology applied. There was also no association between the HGS or TMM and the performance levels of the climbers studied. The FLM values of the left and assistant limbs were higher in elite climbers, giving relevance to the application of DXA for the evaluation of both total and regional body composition. In this sense, future research could focus on a reliable on-field evaluation of FLM to guide trainers in the execution and monitoring of muscle hypertrophy protocols in this body segment. Finally, the upper-body traction strength was significantly higher in elite-level climbers, independent of TMM, highlighting the development of a specific sports skill beyond the upper-body muscle mass.

## Figures and Tables

**Figure 1 jfmk-09-00228-f001:**
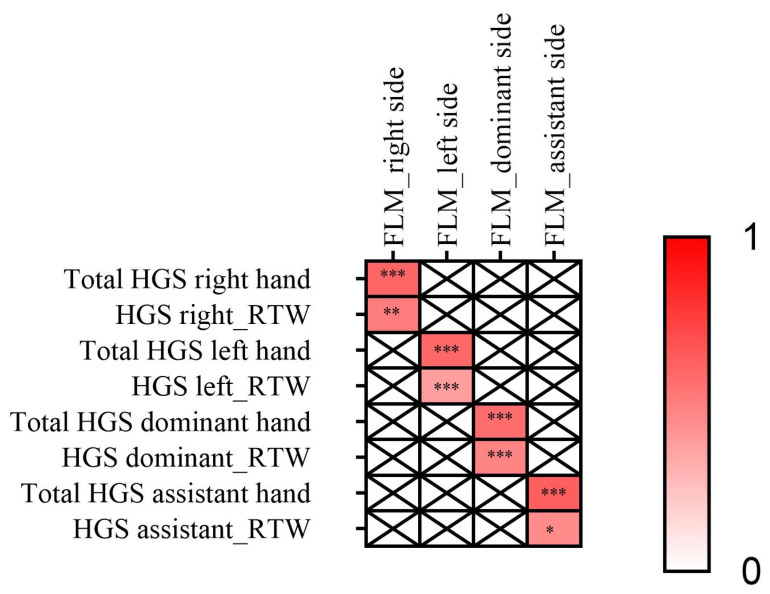
Association between forearm lean mass and maximal hand-grip strength in climbers. Pearson correlation coefficients were determined to examine the association of the right, left, dominant and assistant FLM with the HGS of the right, left, dominant and assistant hands. * *p* ≤ 0.05, ** *p* ≤ 0.01, *** *p* ≤ 0.001. FLM: forearm lean mass; HGS: hand grip strength; RTW: relative to total weight (kg).

**Figure 2 jfmk-09-00228-f002:**
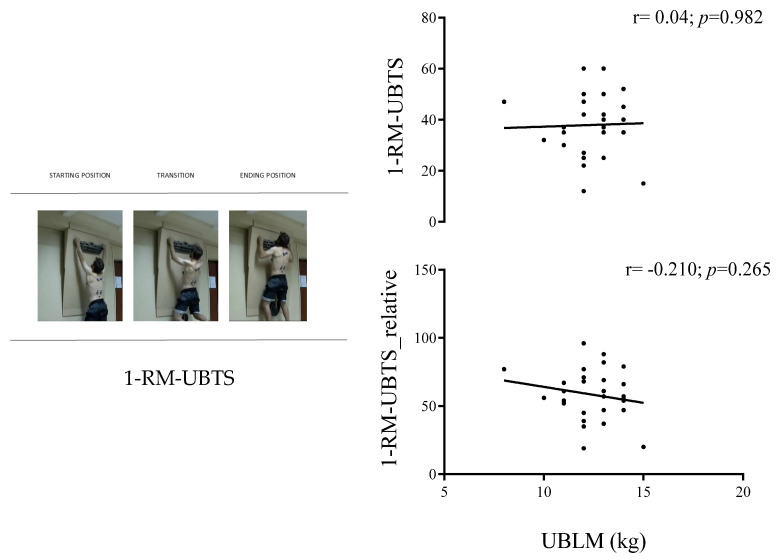
Association between upper-body lean mass and upper-body traction strength in climbers. Pearson correlation coefficients were determined to examine the association of upper-body lean mass (UBLM), absolute and relative, with total weight, with 1-maximum repetition of upper-body traction force (1-RM-UBTS).

**Table 1 jfmk-09-00228-t001:** Anthropometric characteristics and body composition evaluated by DXA in climbers, according to the IRCRA scale.

	Level 1 *n* = 9	Level 2 *n* = 11	Level 3 *n* = 10	*p* Value
Age (y.)	25.3 ± 3.2	26.2 ± 5.9	26.9 ± 5.6	0.803
Weight (kg)	63.5 ± 6.4	65.5 ± 6.0	69.0 ± 6.13	0.155
Height (m)	1.71 ± 0.06	1.73 ± 0.04	1.75 ± 0.07	0.453
BMI (kg/m^2^)	21.6 ± 1.8	21.9 ± 1.5	22.6 ± 1.3	0.358
Total FM (%)	15.3 ± 2.9	14.7 ± 2.9	15.2 ± 2.8	0.691
Total LM (kg)	51.7 ± 4.9	53.5 ± 4.2	56.0 ± 5.6	0.178
TMM (kg)	27.9 ± 3.4	29.0 ± 2.5	30.9 ± 3.7	0.129
, RTW	43.9 ± 1.7	44.3 ± 2.0	44.7 ± 1.7	0.610
FLM (kg), Right side	1.02 ± 0.13	1.09 ± 0.10	1.16 ± 0.17	0.084
, Left side	1.00 ± 0.11 ^a^	1.08 ± 0.10 ^a^	1.16 ± 0.16 ^b^	0.047
, Dominant	1.02 ± 0.13	1.096 ± 0.097	1.17 ± 0.17	0.061
, Assistant	1.00 ± 0.1 ^a^	1.08 ± 0.096 ^a^	1.16 ± 0.169 ^b^	0.041
UBLM (kg)	15.6 ± 1.66	15.5 ± 2.09	16.5 ± 1.31	0.414
, RTW	0.20 ± 0.01	0.19 ± 0.02	0.20 ± 0.01	0.460

Median values and sample sizes are provided. BMI: body mass index; FM: fat mass; LM: lean mass; FLM: forearm lean mass; TMM: total muscle mass; RTW: relative to total weight; UBLM: upper-body lean mass. An ANOVA test was used to compare the anthropometry and body composition across exercise groups. The *p* values are provided. Values in a row with different superscript letters (a vs. b), are significantly different, *p* < 0.05 (Bonferroni’s correction for multiple comparisons).

**Table 2 jfmk-09-00228-t002:** Hand grip strength and upper-body traction strength in climbers, according to the IRCRA scale.

	Level 1 *n* = 9	Level 2 *n* = 11	Level 3 *n* = 10	*p* Value
Right hand (kg)	48.4 ± 7.1	51.1 ± 6.4	55.8 ± 11.2	0.177
, RTW	0.764 ± 0.087	0.786 ± 0.114	0.809 ± 0.147	0.721
, adj. TMM ^#^	49.9 (44.3–55.5)	51.5 (46.5–56.4)	54.1 (48.7–59.4)	0.572
, adj. FLM ^&^	51.0 (45.8–56.3)	51.1 (46.6–55.6)	53.4 (48.5–58.4)	0.734
Left hand (kg)	49.1 ± 6.3	48.6 ± 7.5	54.4 ± 7.9	0.158
, RTW	0.774 ± 0.066	0.746 ± 0.127	0.789 ± 0.099	0.631
, adj. TMM ^#^	50.6 (45.9–55.2)	48.9 (44.8–53.0)	52.7 (48.2–57.2)	0.456
, adj. FLM ^&^	51.6 (47.1–56.1)	48.6 (44.7–52.5)	52.1 (47.8–56.4)	0.392
Dominant hand (kg)	49.1 ± 6.9	51.1 ± 6.4	55.0 ± 10.8	0.300
, RTW	0.773 ± 0.074	0.786 ± 0.114	0.798 ± 0.145	0.903
, adj. TMM ^#^	50.6 (45.2–56.0)	51.5 (46.7–56.2)	53.2 (48.0–58.5)	0.777
, adj. FLM ^&^	51.6 (46.4–56.9)	51.2 (46.7–55.7)	52.6 (47.7–57.6)	0.910
Assistant hand (kg)	48.4 ± 6.5	48.5 ± 7.5	55.2 ± 8.5	0.092
, RTW	0.764 ± 0.080	0.746 ± 0.127	0.800 ± 0.103	0.516
, adj. TMM ^#^	49.9 (45.0–54.8)	48.9 (44.6–53.2)	53.5 (48.8–58.2)	0.337
, adj. FLM ^&^	51.2 (46.5–55.8)	48.7 (44.7–52.6)	52.6 (48.2–57.0)	0.379
1-MR-UBTS (kg)	31.4 ± 9.9	38.2 ± 16.7	44.3 ± 10.2	0.047
, RTW	0.50 ± 0.16	0.59 ± 0.19	0.65 ± 0.17	0.202
, adj. TMM ^#^	30.9 (23.2–38.6)	38.1 (31.3–44.8)	44.8 (37.5–52.2)	0.049
, adj. UBLM *	31.2 (23.8–38.7)	38.0 (31.2–44.8)	44.6 (37.4–51.8)	0.047

Mean ± standard deviation, or mean adjusted by total muscle mass (TMM) ^#^, forearm lean mass (FLM) ^&^, or upper-body lean mass (UBLM) * (95% confidence interval), and sample sizes are provided. 1-MR-UBTS: 1 maximum repetition of upper-body traction strength; RTW: relative to total weight (kg). ANOVA test was used to compare hand grip strength (absolute and RTW). Hand grip strength adjusted mean by partial correlation analysis: significant effect of TMM (*p* < 0.05) and FLM (*p* < 0.005). Upper-body traction strength adjusted: no significant effect of TMM (*p* = 0.588) or UBLM (*p* = 0.627) and significant effect of performance level (Bonferroni post-hoc test: level 1 vs. level 3, *p* = 0.015 and 0.014, respectively).

## Data Availability

Thesis available in Faculty of Medicine library repository.

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
