# Peer review of "Regional Body Composition and Strength, Not Total Body Composition, Are Determinants of Performance in Climbers"

_jfmk, 2024, doi:10.3390/jfmk9040228_

Round 1

Reviewer 1 Report

Comments and Suggestions for Authors

The manuscript titled “Regional body composition and strength are determinants of performance in climbers, but not total body composition” is well written and has scientific soundness. However, I have some comments and suggestions for improving the manuscript.

Please remove full stop from the title.

Abstract

There is no need to introduce measuring units in the abstract. E.g., kg.

Methods

Please specify how did you categorize climbers into these three levels, what was the criterion?

Please explain in more detail how does UBTS look like, what is that standardized movement?

Results

Please specify what superscript letters a and b stand for, in the legend. The p-value is redundant if you are showing significant differences by these superscript letters and explained in the text of the results.

Table 1 and 2: please adjust tables, put units in brackets and expand table that each variable has its row. Also, add the climbing level according to IRCRA scale in Table 1.

Figure 1: It is not clear for me what does the HGS assistant and dominant mean, when you have left and right. Some of these values are redundant.

Why you did not calculate the correlations with climbing level, maybe that could expand the importance of the manuscript and add to the discussion.

Author Response

Reviewer 1

Comments and Suggestions for Authors: The manuscript titled “Regional body composition and strength are determinants of performance in climbers, but not total body composition” is well written and has scientific soundness. However, I have some comments and suggestions for improving the manuscript.

Response: We acknowledge the reviewer for the positive and constructive comments. As proposed, we have deeply considered all the suggestions.

Comment: Please remove full stop from the title.

Response: We thank the Reviewer’s comment. We have removed the full stop on the title.

Comment: Abstract

There is no need to introduce measuring units in the abstract. E.g., kg.

Response: We have removed all the measuring units in the abstract

Comment: Methods

Please specify how did you categorize climbers into these three levels, what was the criterion?

Response: We have added this information in the manuscript, beyond the simple reference of IRCRA scale.

Comment: Please explain in more detail how does UBTS look like, what is that standardized movement?

Response: Thanks for the very constructive comment. In the manuscript, we have incorporated details about the protocol of measurement.

Comment: Results

Please specify what superscript letters a and b stand for, in the legend. The p-value is redundant if you are showing significant differences by these superscript letters and explained in the text of the results.

Response: We thank the Reviewer´s constructive comment. The footnote to Table 1 shows the next statement: “Different letters superscript means significant difference after Bonferroni post hoc test”, and the differences of each letter are not specified. We have modified the footnote and add the next information: Values in a row with different superscript letters (a vs. b), are significantly different, P < 0.05 (Bonfer-roni’s correction for multiple comparisons).

Comment: Table 1 and 2: please adjust tables, put units in brackets and expand table that each variable has its row. Also, add the climbing level according to IRCRA scale in Table 1.

Response: As suggested, we have modified it in the manuscript.

Comment: Figure 1: It is not clear for me what does the HGS assistant and dominant mean, when you have left and right. Some of these values are redundant.

Response: Thank you for this very important comment. The values shown in the figure are not redundant because the dominant hand does not always correspond to the right hand.  It is for this reason that the Pearson correlations between forearm lean mass with these variables were different: right forearm (r = 0.603), left forearm (r = 0.587), dominant (r = 0.571) and assistant (r = 0.624).

Comment: Why you did not calculate the correlations with climbing level, maybe that could expand the importance of the manuscript and add to the discussion.

Response: We thank the Reviewer’s comment. The correlation between interest variables (forearm and upper lean mass) with climbing level could not be done due to this last being a categoric variable. In addition, correlations within each climbing level it is not suitable due to the low number of subjects in each group.

Reviewer 2 Report

Comments and Suggestions for Authors

The sport to which the work is referenced is currently very popular, so it is a topic of interest.

Introduction

It is very short and should differentiate the different types of climbing to understand the needs that are very different according to the discipline.

In the object of study, in addition to the measurement variables considered, the technical novel, which conditions the dependent variables, is not explicitly mentioned.

Methods

The ethics committee code must also be noted.

It is worth demonstrating the proposed exercise in a figure for better understanding.

Discussion

A statistical analysis should be established considering the level of the athlete (not as a segmentation of the sample but as a variable that can influence the result). The level of climbers is influenced by technical or experience differences, which should be reflected in the results.

A section on limitations could be added: the sample was not large, and there was no presence of women.

Author Response

Reviewer 2

Comments and Suggestions for Authors: The sport to which the work is referenced is currently very popular, so it is a topic of interest.

Response: We acknowledge the reviewer for the positive comments.

Comment: Introduction

It is very short and should differentiate the different types of climbing to understand the needs that are very different according to the discipline.

In the object of study, in addition to the measurement variables considered, the technical novel, which conditions the dependent variables, is not explicitly mentioned.

 Response:

Thanks for the very constructive comment. We have added information about different types of climbing and specified in the aims total and regional body composition evaluated by DXA, like the technical novel of this study.

Comment: Methods

The ethics committee code must also be noted.

It is worth demonstrating the proposed exercise in a figure for better understanding.

Response: The number of ethics committees has been added to the manuscript. In addition, we believe that a figure could improve the understanding of upper body traction strength in climbers. In consequence we have added a simple image on the figure 2.

Comment: Discussion

A statistical analysis should be established considering the level of the athlete (not as a segmentation of the sample but as a variable that can influence the result). The level of climbers is influenced by technical or experience differences, which should be reflected in the results.

A section on limitations could be added: the sample was not large, and there was no presence of women.

Response: We thank the Reviewer’s comment. We are aware that the level of the climbers could determine differences in strength and regional hand and upper body composition in climbers, however, the small sample size did not allow us to perform the association analysis taking into account this separation. Likewise, limitations related to sample size and absence of women in the analyses have been highlighted in the discussion.
